# Normative Static Grip Strength of Saudi Arabia’s Population and Influences of Numerous Factors on Grip Strength

**DOI:** 10.3390/healthcare9121647

**Published:** 2021-11-28

**Authors:** Abdalla Alrashdan, Atef M. Ghaleb, Malek Almobarek

**Affiliations:** Department of Industrial Engineering, College of Engineering, Alfaisal University, Riyadh 11451, Saudi Arabia; aalrshdan@alfaisal.edu (A.A.); malmobarek@alfaisal.edu (M.A.)

**Keywords:** static handgrip strength, ergonomics, Jamar dynamometer, muscles, hand force assessment, maximal effort

## Abstract

Most daily tasks require exerting static grip strength which can be challenging for the elderly as their strength diminishes with age. Moreover, normative static grip strength data are important in ergonomics and clinical settings. The goal of this study is to present the gender, age-specific, hand-specific, and body-mass-index-specific handgrip strength reference of Saudi males and females in order to describe the population’s occupational demand and to compare them with the international standards. The secondary objective is to investigate the effects of gender, age group, hand area, and body mass index on the grip strength. A sample of 297 (146 male and 151 female) volunteers aged between 18 and 70 with different occupations participated in the study. Grip strength data were collected using a Jamar dynamometer with standard test position, protocol, and instructions. The mean maximum voluntary grip strength values for males were 38.71 kg and 22.01 kg, respectively. There was a curvilinear relationship of grip strength to age; significant differences between genders, hand area, and some age groups; and a correlation to hand dimensions depending on the gender.

## 1. Introduction

Industrial ergonomics is a subfield of ergonomics concerned with the adaptation of job requirements to the physical requirements of the humans who do tasks. To evaluate population physical abilities and their reference values, populations standards are established. The standards are used for the health, protection, comfort, and efficiency of workers. For manual tasks, equipment and product designs that require grip strength are commonly required to avoid hand injuries. Fatigue or eventual injuries may occur when workers are forced to exert forces that exceed their strength [1,2,3,4,5].

Previous studies offer substantial work using grip strength assessment for different diseases and disorders. The studies demonstrate grip strength (1) as an appropriate measure of overall strength in muscle [6]; (2) as a measure of the impact of various neuromuscular and musculoskeletal disorders [7,8,9] and cardiovascular diseases [10]; (3) as an important factor in the diagnosis of sarcopenia [11] and frailty [12]; and (4) as a predictable outcome such as mortality and potential impairment [13]. Grip strength was recommended for regular use as a vital sign and as a screening method for primary and hospital practice [14,15,16].

To assess grip strength deficiencies, normative reference values are investigated for different populations. Many studies provided normative references in different countries. These include the USA [16], the UK [17], Spain [18], Germany [19], Australia [20], Korea [21], Canada [22], Japan [23], Singapore [24], and Taiwan [25].

Many researchers concluded that several factors may affect grip strength [4,26,27,28,29,30,31,32]. Gender is the most important factor affecting grip strength. Studies reported that the male participants displayed a high grip strength relative to their female counterparts due to the disparity in body structure such as a small muscle mass and high fat mass in females [33,34]. Studies have also shown that grip strength decreases as a person ages [35,36], and there is a close correlation between grip strength and a person’s nutritional status [37,38]. In addition, nutritional status also contributes to certain body mass levels, which were in turn determined to be directly linked to grip strength [39]. Mohan et al. [40] reported that handgrip strength for the dominant hand can be predicted by using hand length and forearm circumference. Fraser et al. [41] and Sirajudeen et al. [42] reported that grip strength and forearm girth have a significant correlation. Incel et al. [43] showed that the strength of the handgrip was higher in groups that dominate the right hand than the left hand. Nevertheless, Reikeras [44] and Roberts et al. [45] have shown that the dominant and non-dominant hand has no significant grip strength difference. Studies show that grip strength at standing posture was more than grip strength at sitting and the supine because of changes in the length of the muscle [46,47,48]. Su et al. [46] argued that a shoulder with 180-degree flexion had a better grip than zero-degree flexion. Swanson et al. [47] showed that the participant’s grip strength was higher in the unsupported arm when compared with the supported arm. Watson and Ring [48] argued that decreasing grip strength is correlated with psychological factors such as depression. Auyeung et al.’s [49] and Choudhary et al.’s [50] studies found connections between the strength of the handgrip and mental health and cognition.

Many studies showed that the strength of the grip differs based on nationality and ethnicity [4,51,52,53,54,55,56,57,58]. Consequently, this study aims to predict the gripping strength of the Saudi population and the factors that affect it. The study investigates the effects of gender, age, body mass index, and hand area on grip strength for the Saudi population. The current study differs from previous research based on its study objective, participants, region, and scope.

## 2. Materials and Methods

### 2.1. Participants

The participants were recruited from Riyadh, the capital of Saudi Arabia. It is the highest populated city and the main financial and government hub of the country. The city is assumed to represent the country’s demographic as it is a mixture of Saudis who moved from different regions to Riyadh seeking job opportunities.

It was hard to recruit participants on a voluntary basis. The participants were solicited by visiting different work facilities such as hospitals, banks, grocery stores, universities, restaurants, households, etc. The randomness of the sample was sought as much as possible. However, some subjects, especially older participants, were selected to complete the data set of all possible categories under study. Inclusion criteria included participants who have not experienced any neuromuscular disease or injury and had no recent or ongoing hand or upper-limb injury. Prior to the study, ethical clearance was obtained from the Institutional Review Board at Alfaisal University (IRB-20037).

The data were collected from both genders between 18 and 70 years old. The sample of each gender is divided into 5 different age groups (18–29, 30–39, 40–49,50–59, and 60–70). A total of 296 subjects participated in the study, which include 146 females and 150 male volunteers. The sample size was determined based on the ISO general standards [59] for establishing anthropometric databases. The following formula was provided to calculate the required sample size that achieve a 95% confidence interval for the 5th and the 95th percentile:(1)$$n\geq {\left(3.006\times \frac{CV}{\alpha }\right)}^{2}\;\mathrm{and}\;CV\;\mathrm{is}\;\mathrm{estimated}\;\mathrm{by}\;\frac{S}{\bar{X}}$$
where *n* is the sample size, *CV* is the coefficient of variation, α is the percentage of the desired relative accuracy, and $\bar{X}$ and *S* are the sample average and standard deviation of the gripping strength, respectively. The required sample size for male and female subjects was determined based on a pilot run of 50 samples of each gender, and the $\bar{X}$ and *S* are calculated for each gender. Using a relative accuracy α = 0.05, the sample size for the males and females were found to be 121 and 67, respectively. The sample size for the study in both genders is greater than the required sample size recommended by ISO standards. The descriptive statistics of the participant data are shown in Table 1 and Table 2.

None of them had self-reported musculoskeletal complaints or problems. All participants were instructed to have a full night’s sleep and to avoid cigarettes and caffeine in the 6 h preceding the tests. Before testing, all participants provided written consent and the protocol and procedures were explained verbally to all participants.

### 2.2. Equipment

The Jamar handgrip dynamometer was used to measure the grip strength with different grip spans for which to accommodate different subjects to enable them to produce their maximal grip strength. The general practice has been the use of the second set of the dynamometer for all or second setting (for females) and third settings (for males). ROSSCRAFT anthropometric tapes and calipers were used for measuring height hand length and width. A digital scale was used to collect the subject’s weight.

### 2.3. Experimental Setup and Procedures

The objectives of the study were shared with the subjects before collecting the data with a full description of the test procedure. Subjects were asked to wear light clothes with bare feet and heads before the scheduled visit to take the measurements. The stature was measured using portable anthropometry from the floor to the head vertex while the subject is standing with a straight erect posture. The weights are recorded from the subjects while standing on the scale with minimum movements and hands alongside the body. The hand length and width are measured using sliding anthropometry, as shown in Figure 1.

Before beginning the data collection, several days were spent to standardize the measurement techniques to reduce data collection variability to a minimum. All subjects participated in a pilot study to familiarize themselves with the gripping strength device, preferred gripping span, and testing procedure. An average temperature of 23.8 °C, relative humidity of 30.6 percent, and continuous lighting were present in the laboratory setting. While the activities were taking place, the experimental site was guaranteed to have no noises or heavy odors. Before the actual test, the participants performed several trial tests at five different grip span settings on the dynamometer to determine their ‘preferred grip span’, for which they felt comfortable and could produce their maximal grip strength. The participants were once again permitted to try the spans one above and one below the preferred span after the initial determination of the preferred span to ensure it that was the one the participant preferred. The Grip strength values were recorded in the morning (9 a.m.–12 p.m.) according to the protocol recommended by the American Society of Hand Therapists [4,60]. This protocol is described as “the sitting position for this recording was sitting in a straight-backed chair with the feet flat on the floor, the shoulder adducted and neutrally rotated, elbow flexed at 90 Degree, and forearm and wrist in a neutral position” as shown in Figure 2.

For all measurements, the arm was free without using any support for it. In order to preserve the normal forearm and wrist positions, the dynamometer has been presented vertically and in alignment with the forearm and the experimenter helped to support the weight of the dynamometer as shown in [4,61]. Collecting the measurement of grip strength follows the protocol suggested by Caldwell et al. (1974) [62]. All the instructions given to the participants were standardized to eliminate subjects’ personal differences. When the participant was ready, he or she was instructed to “increase to maximum exertion (without jerk) in about one second and maintain this effort during a four-second count.

The grip strength measurements were taken 3 times and there was a 5 min time to rest between each reading to provide the subject with sufficient time to recover his strength. The grip strength for a particular subject is approved when the grip strength was within 10 percent for the 3 readings; otherwise, a fourth reading was taken.

### 2.4. Data Analysis

The analysis of the collected data was conducted using IBM SPSS (Statistical Package for Social Sciences) version 23.0. (IBM Corp. Armonk, NY, USA) An analysis of variance (ANOVA with assumptions of normality, independence, and homoscedasticity) was used to investigate the effects of independent factors on grip strength. The variables include gender, age group, hand area, and body mass index. There are two levels of gender (males and females), five levels of age factor (18–29, 30–39, 40–49, 50–59, 60–70 years), three-hand area factor levels (small, medium, and large), and the area was measured by multiplying the hand length (L) by the width (W). The length is measured from the distal wrist crease to the middle of the tipping point of the middle figure. The width is measured from the metacarpal radiale to the metacarpal ulnare, and three levels for the body mass index (underweight, normal, and overweight) were considered. The body mass index stratification was underweight (BMI < 18.5), normal (18.5 ≤ BMI < 25), and overweight (BMI ≥ 25) [63]. The levels of the area factor were found using the K-means clustering algorithm for both genders: for males, small (Area < 146), medium (146 ≤ Area < 176), and large (Area ≥ 176) and, for females, small (Area < 101), medium (101 < Area ≤ 126), and large (Area ≥ 126). The Shapiro–Wilk test was implemented to test data normality [64]. The statistical significance was set at a confidence level of 95%. Accordingly, to investigate the relationship between Weight, Height, BMI, and Hand Area, Pearson Correlation Test with magnitudes of the associations (r = ±0.10–±0.29: small; r = ±0.30–±0.49: moderate; r = ±0.50–±1.0: strong) was conducted; Table 3 shows the Pearson’s correlation coefficients.

## 3. Results

### 3.1. Effect of the Gender

The results show that gender has a significant effect on grip strength (F = 778.518, *p* < 0.001). The male grip strength was significantly higher (mean (SD) = 38.71 (6.08)) when compared with female grip strength (mean (SD) = 22.01 (4.03)), as shown in Figure 3.

### 3.2. Effect of the Age Group

#### 3.2.1. For Male

The statistical analysis showed that there is a significant effect for the age group on the grip strength (F = 7.36, *p* < 0.001). The post hoc analysis shows that the (60–70) age group is the only group significantly weaker than (18–29), (30–39), (40–49), and (50–59) age groups. Additionally, the age group of (30–39) is the strongest followed by (18–29) then (40–49), as shown in Figure 4, but there were no significant differences between the (18–29), (30–39), (40–49), and (50–59) age groups. Table 4 shows the mean and standard deviation of grip strength for age groups of males.

#### 3.2.2. For Female

The statistical analysis showed that there is a significant effect for the age group on the grip strength (F = 5.87, *p* < 0.0001). The post hoc test shows that (60–70) and (50–59) age groups are significantly weaker than the (18–29) and (30–39) age groups. In addition, the (50–59) age group was significantly lower than the (40–49) age group. In addition, the age group of (18–29) is the strongest followed by (30–39), then (40–49), as shown in Figure 5, but there were no significant differences between the (18–29), (30–39), and (40–49) age groups and between the (50–59) and (60–70) age groups. Table 5 shows the mean and standard deviation of grip strength for age groups of females.

### 3.3. Effect of the Hand Area

#### 3.3.1. For Male

The hand area had significantly affected the grip strength F = 3.56, *p* < 0.031. The statistical analysis shows that the grip strength for large hand areas was significantly higher (mean (SD) = 40.47 (4.85)) than for medium hand area (mean (SD) = 38.08 (6.01)) and small hand areas (mean (SD) = 37.41 (7.02)). However, there were no significant differences between grip strength for medium hand areas and small hand areas.

#### 3.3.2. For Female

For female data, the statistical analysis showed that no significant effect for hand area on grip strength. Although the medium hand area is the strongest followed by large hand area then small hand area.

### 3.4. Effect of the Body Mass Index

The results show that the body mass index has no significant effect on the grip strength for males and females.

### 3.5. A Comparison of Average Grip Strength in Saudi Arabia with Other Countries

A comparison of Saudi Arabia’s average grip strength with different countries’ populations according to age groups from prior studies is shown in Figure 6 and Figure 7. As can be seen, there is no significant difference in the average grip strength between Saudi Arabian adults and the Taiwanese population. On the contrary, the grip strength in other societies, such as the United States, the United Kingdom, Turkey, Iran, Switzerland, Australia, and a common database of Western countries, is significantly different in all age groups, as the population of these countries has a higher average grip strength in both sexes than the Saudi population.

## 4. Discussion

The primary purpose of this research was to study the influence of several factors such as gender, age, hand area, and BMI on the grip strength in the Saudi Arabian population. The hypotheses of this study stated that gender, age, hand area, and BMI have a significant effect on grip strength.

In general, the findings of this research are in agreement with a study performed in some other populations. Grip strength has been found to be related to gender, age group, hand area, and body mass index.

### 4.1. Gender Effect

Overall, the grip strength of males has been shown to be considerably higher than that of females. The present research for the population of Saudi Arabia has confirmed this fact again through several earlier studies [4,65]. The strength of muscles is a function of the size of the associated muscles. Since males normally have larger arm muscles and are more active than females in activities involving strength, their mean grip strength is estimated to be higher than females [4].

### 4.2. Age Group Effect

In males, there was no inverse or direct relationship between grip strength and age group, while in females, the relationship is inverse, as the age increases, the grip strength decreases. The strength of the male grip increases and peaks somewhere between 30 and 39 years of age and gradually decreases until about 49 years of age and then begins to decrease significantly. This relationship is similar to what has been reported in previous studies [4,56]. However, the grip strength of females tends to decrease as they get older. It is possible that the difference in the length of time required to maintain grip strength between men and women may be partially due to genetics and partly due to resistance to age-related muscle loss because people maintain arm muscle mass by engaging in physical activities for a longer period of time, and thus fighting age-related muscle loss for a longer period of time [4]. Grip strength peaks during the third and fourth decades and declines thereafter as reported by many researchers [4,25,56,57]. The results for males in this study are completely consistent with those studies. For example, Eksioglu 2016 [4] reported that male grip strength peaks within the 30–39 year age group and regularly decrease thereafter. In contrast, female grip strength peaks within the 18–29-year age group and then decrease as the age group increase and this finding is similar to our findings.

### 4.3. Hand Area Effect

The hand area had significantly affected the grip strength for males but did not affect the grip strength for females. It was hard to compare our results regarding hand area since there was no existence of a similar study available in the literature. However, ultrasound can be used to determine the cross-sectional area of certain hand muscles. As a result, this technique may be beneficial for monitoring muscle reinnervation in patients with peripheral nerve damage, complementing strength dynamometers [66].

In additional hands, exert gripping force on the object to gain control and prevent it from falling. The total force is found by integrating the shear force acting on the logididunal contact area between the hand surface and the gripped object. The higher the contact area the bigger the gripping force. It was found that the contact area is correlated with hand area [34].

Finally, the total gripping force is the sum of all individual forces acting by hand anatomical components. Considering the total area will include the anatomical parts of the hand.

### 4.4. BMI Effect

When body mass index was classified into underweight, normal, and overweight, we found that there was no correlation between body mass index with grip strength for males or females. These results are similar to those results found by many researchers [67,68,69]. In contrast, Massy-Westropp et al. 2011 [20] concluded that there was a very small positive correlation between body mass index and grip strength in the youngest and oldest age groups in the sample. They also noted that body mass index was negatively correlated with grip strength in the (40–49), (50–59), and (60–65, 67–70) age groups. In additional, Dhananjaya et al. 2017 [70] noted a significant negative correlation between body mass index and grip strength.

### 4.5. Grip Strength in Different Populations

As a matter of fact, handgrip strength comparisons between regions may shed light on historical regional variances in genetic variables, dietary inadequacies, and/or sociocultural environments [71]. When the grip strength of Saudi adults (of any gender) was compared to the grip strength of Taiwan adults, no significant differences were found. However, grip strength in other societies, such as the United States, the United Kingdom, Turkey, Iran, Switzerland, Australia, and a common database of Western countries, differ significantly in all age groups, as shown in Figure 5 and Figure 6. The population of these countries has a higher average grip strength in both sexes than the Saudi population. According to Kamarul et al. and Xiao et al., there is a considerable difference in grip strength between Malaysian adults and Chinese adults with Western dominance [55,72]. The likelihood is that the variations in grip strength normative data values are due in large part to the disparity in anthropometric parameters. Physical characteristics such as height and weight depend on the type of race [73]. As a result, the vast disparity in grip strength between Saudi and Taiwanese populations and Western, Turkish, and Iranian [74] populations is due to racial differences. The influence of race can be so strong that it can overshadow the influence of other important factors such as diet, socioeconomic status, and cultural group. The significance of the race impact was demonstrated in a study conducted by Anjum et al., in which the geographical and socio-economic variables of the comparable populations were examined. It was discovered that British individuals had a statistically significant dominance in all grip strengths when compared to Asian people among the residents of West Yorkshire, England, which had a predominantly British population and a large proportion of Asian people [75].

As gripping strength has been linked to anthropometric and socioeconomic factors of a particular society, Saudi Arabia is no different from other developing countries when compared with industrialized countries concerning body size and socioeconomic factors. A study by Alrashdan et al. [76] has shown that the weight and stature of US females and some Western countries are significantly higher than Saudi females. At the socioeconomic level, the Saudi health-based socioeconomic indices are lower compared to the US and Western countries due to the lack of education and healthcare services provided to rural communities living far from educational institutions or healthcare facilities. These factors might explain the smaller gripping strength of the Saudi population compared to Western countries, as well as the nature of the work performed by the Saudi population, as most of them work in office work, which does not require physical work. In addition, fitness here is less than in other countries, and obesity in the Kingdom of Saudi Arabia is greater than that in other countries; all of these factors helped to make the grip strength of the Saudis less.

## 5. Limitations

The primary limitation of our study is that, while the analysis takes into account characteristics such as age, gender, hand area, and body mass index, it does not take into consideration anthropometric factors (for example, wrist circumference), level of physical activity of the participants, and socioeconomic status that have been shown to influence grip strength in previous research. The study also has another limitation in that it only gives data for dominant hands rather than data for both dominant and non-dominant hands. A third limitation of our study is that it was only undertaken in one city, which is the capital, although the capital usually contains residents representing all parts of the country due to the availability of services and job opportunities in the capital.

## 6. Conclusions

In this study, the hand-grip strength norms of healthy adult Saudi citizens aged 18 to 70 were established. The data may serve as reference values in designing for grip strength (tasks, equipment, and consumer products). The availability of such data makes it possible to create biomechanical models and develop ergonomics evaluation tools. Additionally, the results of this research may also be helpful for treating clients and for preparing them for a job return.

These findings show how grip strength for Saudi Arabia’s population differs from those of other populations. This study provides new information on the various factors that influence grip strength. An overwhelming majority of these variables have already been studied in other populations. Despite this, the results for the study population demonstrate the impact of these factors on grip strength.

## Figures and Tables

**Figure 1 healthcare-09-01647-f001:**
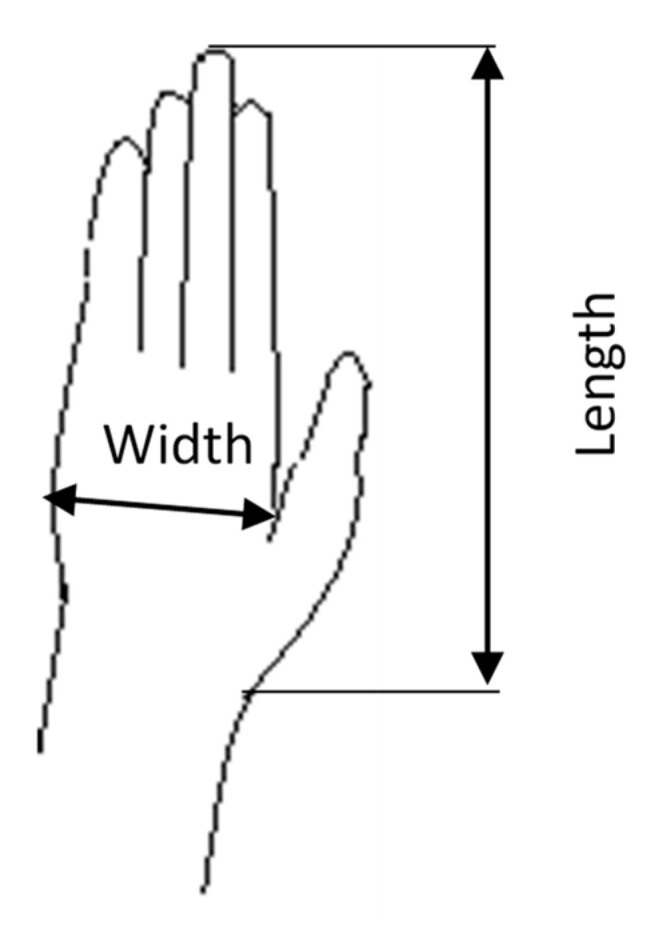
Hand area.

**Figure 2 healthcare-09-01647-f002:**
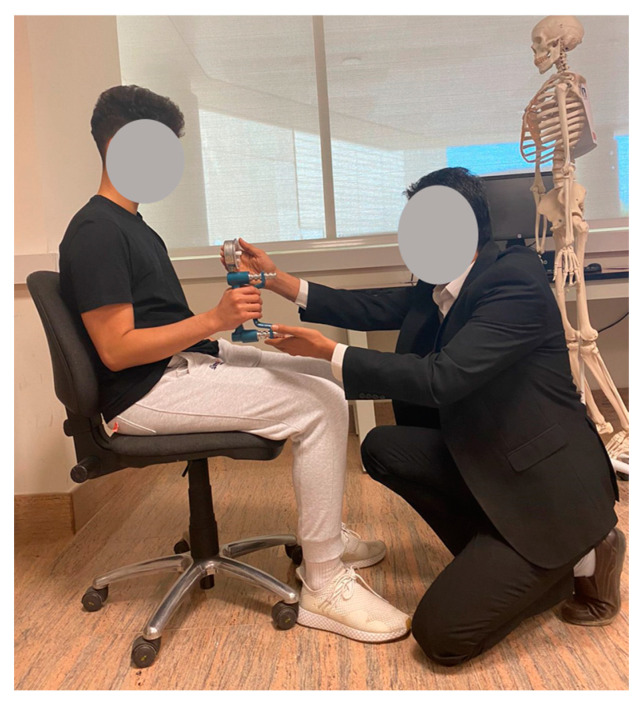
Illustration of the position for the measurement of Grip Strength.

**Figure 3 healthcare-09-01647-f003:**
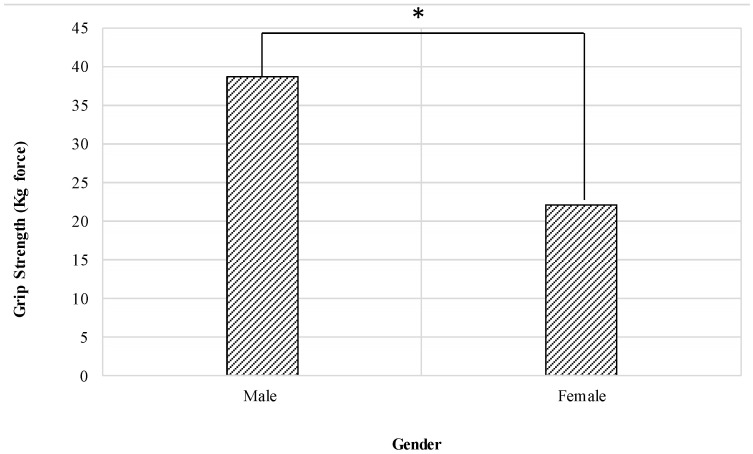
Effect of gender on the grip strength (* means there is a significant difference between levels).

**Figure 4 healthcare-09-01647-f004:**
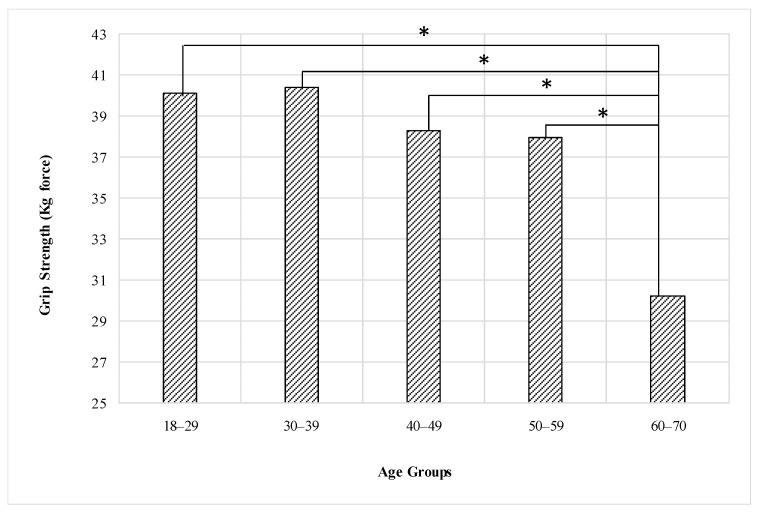
Effect of the age group on the grip strength of the male (* means there is a significant difference between levels).

**Figure 5 healthcare-09-01647-f005:**
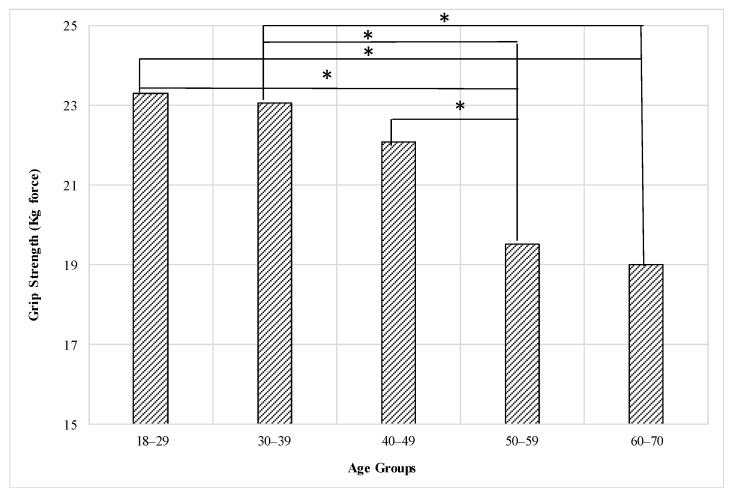
Effect of the age group on the grip strength of females (* means there is a significant difference between levels).

**Figure 6 healthcare-09-01647-f006:**
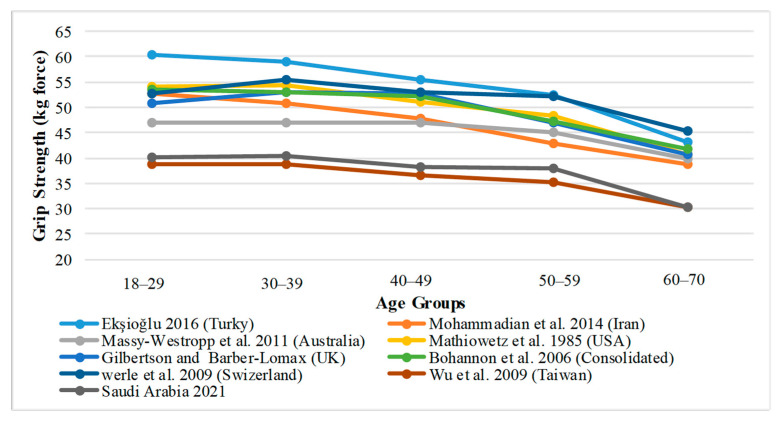
Male, regional reports of mean grip strength by age groups.

**Figure 7 healthcare-09-01647-f007:**
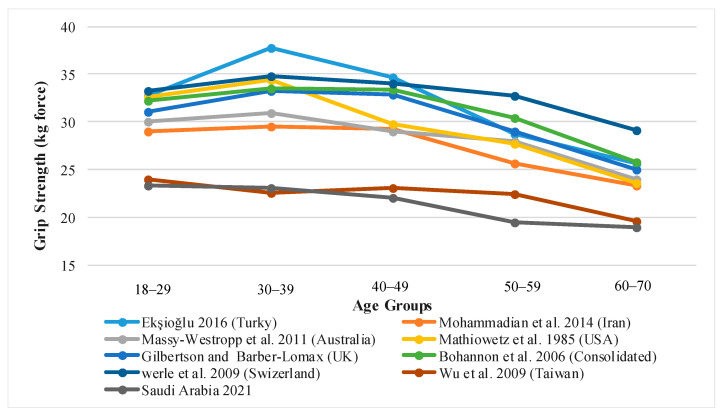
Female, regional reports of mean grip strength by age groups.

**Table 1 healthcare-09-01647-t001:** Describes participant information.

Body Measurement	Male (*n* = 146)	Female (*n* = 150)
	Mean ± SD	Mean ± SD
Age (year)	38.5 ± 12.6	38.7 ± 12.9
Weight (kg)	85.5 ± 15.5	66.1 ± 11.8
Hight (cm)	175.7 ± 6.3	161.8 ± 5.4
BMI (kg/m^2^)	27.7 ± 4.7	25.2 ± 4.5
Hand length (cm)	18.4 ± 1.3	15.9 ± 1.4
Hand width (cm)	8.8 ± 1.2	7.3 ± 0.9
Hand area (cm^2^)	163.6 ± 27.5	116.8 ± 20.3

**Table 2 healthcare-09-01647-t002:** Describes participant information.

Age Group	No. of Participants		Height (cm)		Weight (kg)		BMI (kg/m^2^)		Hand Area (cm^2^)	
	Male	Female	Male	Female	Male	Female	Male	Female	Male	Female
18–29	30	30	175.6	159.75	81.1	56.3	26.3	22.04	158.3	141.5
30–39	30	30	177.3	164	84.5	61.3	26.9	27.2	162	106.8
40–49	30	30	177	163.5	85.5	73.2	27.3	27.5	165.2	104.9
50–59	30	30	177.8	161.5	91.1	81.5	28.8	31.3	167	126
60–70	30	26	171	160.25	85.2	58.2	29.1	17.96	165.4	104.2

**Table 3 healthcare-09-01647-t003:** The correlation between weight, height, BMI, and hand area.

Factors	Weight (kg)		Height (cm)		BMI (kg/m^2^)	
	Male	Female	Male	Female	Male	Female
Height	0.298 **	0.168				
BMI	0.901 **	0.933 **	−0.143	−0.192		
Hand area	0.114	−0.103	0.091	−0.059	0.069	−0.084

** Correlation is significant.

**Table 4 healthcare-09-01647-t004:** Summary of males’ grip strength results.

Age Group	Grip Strength (kg Force)	
	Mean	Std. Deviation
18–29	40.09	5.758
30–39	40.42	5.930
40–49	38.29	5.789
50–59	37.95	5.455
60–70	30.2	2.251

**Table 5 healthcare-09-01647-t005:** Summary of females’ grip strength results.

Age Group	Grip Strength (kg Force)	
	Mean	SD
18–29	23.29	4.255
30–39	23.05	3.634
40–49	22.08	4.425
50–59	19.53	2.488
60–70	19	2.309

## Data Availability

All data are available in this research.

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
