# Peer review of "Normative Static Grip Strength of Saudi Arabia’s Population and Influences of Numerous Factors on Grip Strength"

_healthcare, 2021, doi:10.3390/healthcare9121647_

Round 1
Reviewer 1 Report
The authors investigated the normative static grip strength of Saudi Arabia’s population and evaluated some related factors on grip strength. I think that it is important to assess the reference values for grip strength that may serve as comparative data for the patients with hand disorders. However, there are some problems with the methodology.
Major comments
Title
Accurately reflects the content of the manuscript.
Abstract
It can be understood without the content of the manuscript.
Introduction
Adequate and relevant.
Experimental Setup and Procedures
・How did you calculate the hand area? Please describe the methodology
in detail.
Result
・The number of samples and the mean value of height and weight and BMI and hand area of each different group was not found in male and female groups. The authors should clarify these data.
・I think that hand area was associated the height and weight. Did the authors assess the relationship between a hand area and height and weight?
If the authors use the hand area as an evaluation item of the handgrip, you should use the values corrected by height and weight.
Discussion
8 page: line 244-246
・I think that it is original to assess the hand area in this study. However, the authors did not make a deeper analysis of their findings. Why did they evaluate the hand area in this study? I would recommend some more comments on how you would explain the association between the grip strength and hand area in the point of biomechanical and anatomical view.
9 page: line 256-278
・The authors discuss the disparity of handgrip values around the world. The grip strength of Saudi Arabia’s population similar to Taiwan’s result was smaller than those of other countries.
I would recommend making a deep analysis about the smaller grip strength compared with those of other countries and to discuss the factors related to the smaller grip in Saudi Arabia’s population at least.
Reviewer 2 Report
The article is interesting, covering the measurement of an important marker in a specific population in relation to others. However, few points should be addressed by the authors.
Abstract session: Please double check your manuscript before submission there are some typos. Globally the abstract is clear and well written.
Abstract line 9 typo: import (important?)
Abstract line 13 typo: of the gender,….
Introduction session: Again, some typos plus some sentence should be rewritten.
Line 24-25: the sentences is a copy paste of Sanchez 2014. Please consider reformulating it.
Line 47: Please consider using other terms rather than high fat weight (e.g., high fat mass or percentage)
Results:
Please indicate in the bar graphs the symbols of significant or non-significant differences.
Discussions: again, typos. Some sentences need to be better explained.
Line 233-237: please reference your speculation/hypotheses.
Please indicate strength and weaknesses of your research (included limitations).
Reviewer 3 Report
In summary, this is a study that investigates grip strength in the Saudia Arabia's population. The authors analyzed how several factors such as gender, age, hand area, and BMI affected the grip strength. While similar studies have published in other populations, this study was novel in that it investigated a new population. However, the manuscript needs much improvement for the consideration for publication. Especially, grammar, style, and consistency of using capital letters need to be checked. Also, the major concern is whether the authors acquired ethical approval from the IRB.
Introduction: Please check for grammar and typos.
Methods: Did the authors have ethical approval from institutional review board?
Line 78-79: were participants compensated for the participation of the study?
Line 89: did the authors measured the level of physical activity of the participants? this could potentially confound the results. Also, the authors mention later in the discussion about socioeconomic status. did the authors measure any income, education, etc to gauge socioeconomic status? were the population homogenous in terms of ethnicity?
Line 96: what do you mean by all students? where did students come from? how did you ensure that these students voluntarily participate in this study?
Line 114: using the length and width, how was the hand area measured?
Line 131: it might be helpful if the authors include the photo of how the participants were positioned
Results:
Line 205: any formal statistical analysis to compare the results among countries?
Discussion: needs to be better organized. The first paragraph can be used to summarize the key findings and what is novel about this study.
It seems that broad ideas in the discussion are: 1) gender difference in grip strength, 2) age difference in grip strength; 3) association between hand area and grip strength; 4) association between BMI and grip strength; 5) grip strength in different populations. Consider making subsections accordingly.
Don't just list whether the findings are consistent or not consistent with previous studies. Try to make meaningful discussion and what is new and what is different in this study.
Also, the discussion is missing the limitation.
Line 224-226: reference?
Line 228: was there statistical evidence of linear or nonlinear relationships?
Line 233-236: reference?
Line 244-246: even though there may not be exactly similar study available, the authors can still discuss cross sectional area and muscle strength such as the following article to make some explanations for this finding: Mohseny et al. "Ultrasonographic Quantification of Intrinsic Hand Muscle Cross-Sectional Area; Reliability and Validity for Predicting Muscle Strength"
Line 256-258: This first sentence has too many ideas that are not supported by the rest of the paragraph. This paragraph seems incoherent.
Line 259-266: did the authors perform any statistical analysis to support this? or were they just estimating?
Line 280-284: This paragraph rather seems like a clinical implication of this study that could be used in the discussion session rather than conclusion.
Round 2
Reviewer 1 Report
The manuscript has been partially revised. according to reviewers. I think the authors need some work and revise the manuscript.
Results:
・I recommended that the authors should clarify the mean value of the parameter (height, weight, BMI, hand area) and the number of members in each group by age. However, the authors did not respond to my suggestion. The authors should summarize these results in a Table.
・The result that people with larger bodies and hands have greater grip strength seems natural. So I recommended that you should use the values of hand area corrected by height or weight or BMI. If it is difficult to calculate it, the authors should at least show the correlations with hand area and height and weight and BMI.
Discussion
・Line 301-302
It was found that the contact area is correlated with hand 300 area (Investigation of Grip Force, Normal Force, Contact Area, Hand Size, and Handle Size 301 for Cylindrical Handles [34].
→Please delete “(Investigation of Grip Force, Normal Force, Contact Area, Hand Size, and Handle Size 301 for Cylindrical Handles”.
Author Response
Dear Professor
Thank you very much for your compliment.
Please find the attached file

Reviewer 2 Report
The authors improved their manuscript following the suggestions from the previous revision.
Although some typos are still detectable (not so important for the context but is just a matter of double reading your work and present a better version and easier to read), the authors improved all sections of the manuscript providing more and clearer information and by strongly supporting their statements in the discussion section.
Good luck for the next steps of the revision process.
Author Response
Dear Professor
Thank you very much for your compliment
Reviewer 3 Report
Limitations should be presented before conclusion.
Author Response
Dear Professor
Thank you very much for your compliment.
Done, we presente the limitations before conclusion.
Regards,